# Measuring leadership an assessment of the Multifactor Leadership Questionnaire

**Joan Manuel Batista-Foguet**[1]*, **Marc Esteve**[2,3], **Arjen van Witteloostuijn**[4]

**1** Department of People Management and Organisation, ESADE Business School, Ramon Llull University, Barcelona, Spain, **2** School of Public Policy, University College London, London, United Kingdom, **3** Department of Society, Politics and Sustainability, ESADE Business School - Ramon Llull University, Barcelona, Spain, **4** School of Business and Economics, Vrije University Amsterdam, Amsterdam, Netherlands

\* joanm.batista@esade.edu

## Abstract

Although the most used measure of transformational leadership, the Multifactor Leadership Questionnaire (MLQ), has been the subject of intense scrutiny among leadership scholars, little interest has been shown in analyzing the relationship between its underlying constructs and / or their measures. The present study identifies a formative factor structure for most MLQ first-order factors, replacing the usual reflective model. We demonstrate the value of this structure using data from two different samples. First, we applied the MLQ to a sample of 129 police officers from the Catalan Police workforce. Second, we ran an online survey with 300 US citizens. We argue that three second-order factors (transformational, transactional, and laissez faire) should be used as emergent aggregate multidimensional models to describe three different leadership styles, challenging the ubiquitous multidimensional latent models favored in the extant literature. We then propose that transformational/charismatic leadership should be treated as a multidimensional emergent profile model, replacing the leadership development order of precedence, which is dominant in modern leadership research.

## Introduction

The Multifactor Leadership Questionnaire (MLQ) is a leading survey instrument used to assess leadership [1, 2]. Created to test Bass and Avolio's model of leadership [3], the MLQ has been used in many studies to examine general leadership theory, including many individual and organizational correlates in the context of the Organizational Behavior domain. Initially known as the MLQ-5X, its strength depends on its ability to capture several leadership styles in a single measurement instrument [4, 5]. It incorporates a range of nine scales: five to capture transformational, three to measure transactional, and one to reflect laissez-faire leadership styles, as well as three to include leadership outcomes. Previous studies explored the MLQ's psychometric properties [6, 7]. Its effectiveness and validity have been confirmed in several meta-analyses, e.g. [2, 8]. By now, the MLQ can be considered to be a prominent example of a clear academic (and commercial) success.

**Data Availability Statement:** All relevant data are within the paper and its S1 Data and S1 and S2 Files.

**Funding:** Agència de Gestió d'Ajuts Universitaris i de Recerca, Grant/Award Number: SGR Program,

2017-SGR-1556; Ministerio de Economía y
Competitividad, Grant/Award Number: CSO2016-
80823-P.

**Competing interests:** The authors have declared
that no competing interests exist.

Although recognized as an example of scientific success [9], many researchers have criticized the instrument, presenting evidence to suggest that its "full-range theory" is not as "full" as initially suggested [10–12]. Critiques of the MLQ note the following potential flaws: (1) its questionable multidimensional structure; (2) its lack of connection with theory; and (3) the doubtful way in which different sub-dimensions of the MLQ combine to form a unitary model. Hunt [1] and Yukl [13] were among the first to raise concerns about the multidimensional structure of the MLQ. Yukl [13] argued that the MLQ ignores important transformational behaviors displayed by leaders, including inspiring, empowering, and developing behaviors. Hunt [1] pointed out that the two-factor model has limitations by failing to consider important contextual variables that could affect leadership and its consequences. In a similar vein, Judge and Piccolo [8] warned that the MLQ has an "elusive" multidimensional structure. "A construct is defined as multidimensional when it refers to several distinct but related dimensions that can be connected parsimoniously and meaningfully into one single holistic concept" ([14] p. 803) [15]. According to van Knippenberg and Sitkin [12], inferences cannot be drawn from Transformational-Charismatic Leadership's (TCL) unidimensionality about leadership as a unitary construct, given the lack of discriminant validity. This is especially true when the unitary structure of higher-order constructs is defined by a theory of multidimensional nature.

Other researchers raised validity concerns because the TCL sub-dimensions are combined into a unitary model without explicit justification [16–18]. In a similar vein, Wong, Law, and Huang [15] exposed this issue by arguing that it is common in management studies to use "a general label or umbrella to refer to a group of inter-related constructs and assume that the label is a multidimensional construct. Like them, we refer to these labels as "pseudo-multidimensional constructs" ([15] p. 745). They criticize the lack of definitions linking the overall multidimensional construct with each of its sub-dimensions.

Generally, methodological research in Organizational Behavior warned against problems associated with multidimensional constructs, and proposed alternatives [15, 19, 20]). This stream of work appears to be disconnected from leadership research, although earlier studies challenged researchers to evaluate critically the psychometric properties of extant leadership measures [11–13, 21–23]. This paper bridges these two bodies of literature by developing a critical assessment of the MLQ specifically, and proposing ways to test multidimensional leadership constructs generally.

Specifically, we identify a formative factor structure for most MLQ first-order factors, replacing the usual reflective model. We demonstrate the value of this structure using data from two different datasets sampled in two different countries. We argue that three second-order factors (transformational, transactional, and laissez faire) should be used as multidimensional emergent models to describe three different leadership styles, challenging the ubiquitous multidimensional latent models favored in the extant literature. We then propose that transformational/charismatic leadership should be treated as a multidimensional emergent profile model, replacing the leadership development order of precedence (ranging from "laissez faire" to "transformational"), which has dominated leadership research in recent decades.

## Background

It appears that the original underlying methodological assumptions, based on Classical Test Theory (CTT) [24], have never been questioned in the context of the MLQ (or other leadership measures, for that matter), although the nature of the associated full-range theory as a multidimensional construct has been discussed. The present study addresses the gap between measurement and the conceptual model discussed by van Knippenberg and Sitkin [12] by testing

whether a few of the identified inconsistencies can be attributed to the assumed factorial structure of the MLQ-5X.

The construct validity of the MLQ has attracted the interest of many scholars who use survey data to analyze leadership. In fact, the original and subsequent refinements of the MLQ-5X have always been framed within CTT. In the original Bass study [25] "a principal component factor analysis was run with varimax rotation, for the data from 104 military officers" (p. 207). At a later stage of development, many studies framed the MLQ similarly within CTT (e.g., [5, 6, 23, 26–31]. Consequently, factor analysis has oftentimes been applied to prove the MLQ's convergent and discriminant validity, eventually searching for an "underlying single common factor" to account for the inter-correlations among items. To assess reliability, Cronbach's alpha and alternative approaches are used, all based on item consistency. Several recent studies related the MLQ-5X dimensions to other constructs, assuming the factorial structure originally proposed by Bass and Avolio (e.g., [32–34]).

Van Knippenberg and Sitkin [12] have shown that this state of affairs is not unique to the MLQ, referring to other well-known leadership measures, including the Conger–Kanungo Scale, the measure developed by Podsakoff et al. [17], and the scale introduced by De Hoogh and den Hartog [35] and De Hoogh et al. [36]. Other measures were unidimensional to begin with, despite being based on multidimensional conceptualizations [37]. Van Knippenberg and Sitkin [12] argue that the multidimensional structure of the MLQ is simply assumed, pointing out the lack of epistemic theory underpinning the MLQ, as there is "no theoretically grounded configurational model to explain how the different dimensions combine to form charismatic–transformational leadership" (p. 45).

The present study sets out to find evidence of construct validity of the MLQ-5X, taking this well-established case as an illustrative and highly prominent example of the broader issue of the weak psychometric foundation of leadership scales. The AERA, APA, and NCME Standards for Educational and Psychological Testing [38] define validity as "the degree to which evidence and theory support the interpretations of test scores entailed by proposed uses of tests." This "evidence" can be based on test content, response processes, internal structure, relations with other variables, convergent and discriminant evidence, relationships between test criteria, validity generalization, and test-based evidence. We argue that the internal structure of the MLQ is undermined by a lack of evidence.

Basically, rather surprisingly, researchers have never tested empirically the extent to which relationships between items in the survey and survey components constitute construct-based leadership scores. Major threats to construct validity include tacit assumptions about the MLQ structure—i.e., the unidimensionality of first-order factors, and the nature of leadership styles as multidimensional constructs. This paper explores the consequences of these threats for the assessment of the survey's measurement quality. We argue against the assumed factorial structure derived from the CTT framework. Following Law, Wong, and Mobley's [39] proposed taxonomy of multidimensional constructs, we discuss the nature of three higher-order factors: transformational, transactional, and laissez faire leadership styles. We then propose a better way of relating these three multidimensional constructs to provide a full range of leadership characterizations.

## Epistemology

In its current form, the MLQ-5X is designed to measure nine leadership factors. At the time we collected our data, the first five (idealized influence attributes, idealized influence behaviors, inspirational motivation, intellectual stimulation, and individualized consideration) are assumed to capture transformational leadership. The next three (contingent rewards, active

management-by-exception, and passive management-by-exception) are claimed to be associated with transactional leadership. The last factor is said to be concerned with laissez faire or non-leadership. In the latest version of the MLQ, these factors have been relabeled into "builds trust", "acts with integrity", "encourages others", "encourages innovative thinking", coaches and develops people", "rewards achievement", "monitors deviations and mistakes", fights fires", and "avoids involvement". This is not essential for what we do in the current paper.

To assess the factorial structure of the MLQ, we begin by clarifying the conceptual specifications of all indicators, to allow for deciding whether the indicators form a *formative* or *reflective* model.

In the last century or so, indicators were oftentimes assumed to represent reflections of a particular construct. That is, they were used as reflective (or effect) indicators to operationalize that specific construct. Even constructs inadvertently proposed as formative have, for decades, been analyzed as reflective. For instance, this happened with the socio-economic status construct across a wide range of different disciplines (e.g., [14, 40–44]).

Why did this happen, given that many methodologists, from Fornell and Bookstein [40] and Bollen [45–47], and Hardin [48], have shown the need for alternative factor specifications based on formative (or cause) indicators? Instead of reflecting the underlying construct, indicators represent facets of the construct, which is actually formed from its indicators [49]. The main difference between formative and reflective models is that formative models combine measures to form weighted linear composites that represent theoretical concepts, while reflective models treat measures as outcomes of unobserved latent variables [50]. The distinction between reflective and formative models is particularly important when assessing the dimensionality of a construct. Reflective models "are assumed to represent a single dimension, such as that the measures describe the same underlying construct, and each measure is designed to capture the construct in its entirely" ([50] p. 373). Reflective indicators are actually samples of the possible observable indicators; they are thus interchangeable.

By contrast, a formative model is typically described using different dimensions of a construct; if one measure is eliminated, the construct is incomplete, as the remaining construct measures will not capture those construct facets [43, 51]. Formative indicators specify that the factor is a linear combination of contributor indicators plus a disturbance, which means that causal indicators cannot completely determine the latent variable. As the composite indicators perfectly determine the construct, the disturbance is zero. Since there is a certain ambiguity in the term formative indicators, they are synonymous with causal indicators in the present paper, and differentiated from composite indicators, which may not have conceptual unity and can be arbitrary combinations of observable variables [52].

In the conceptual specification, it is essential to determine the nature and direction of the relationships between a construct and its indicators. The conceptual specification of the full-range theory requires that epistemic relationships must be justified on conceptual grounds, before any empirics are introduced [53]. Consequently, it is crucial to clarify which of the nine MLQ-underlying dimensions are manifested through a series of indicators, using a *reflective* model. Since reflective items are conceptualized as manifestations of the assumed unidimensional latent construct, causality flows from the construct to the items, as visualized in Fig 1.

Several indicators in the MLQ are reflective items, such as in the transformational dimension *Inspirational Motivation* (*IM*). If a leader has the skills needed to inspire others, s/he should manifest behaviors such as those described by the four items that refer to the confidence provided by the leader's expressions of optimism about the future vision. Therefore, these reflective indicators are expected to covary, and their covariation is attributed to their common source (i.e., *IM* leadership skills). Defining *IM* as a reflective factor model implies that the four indicators are essentially samples of interchangeable possible manifestations

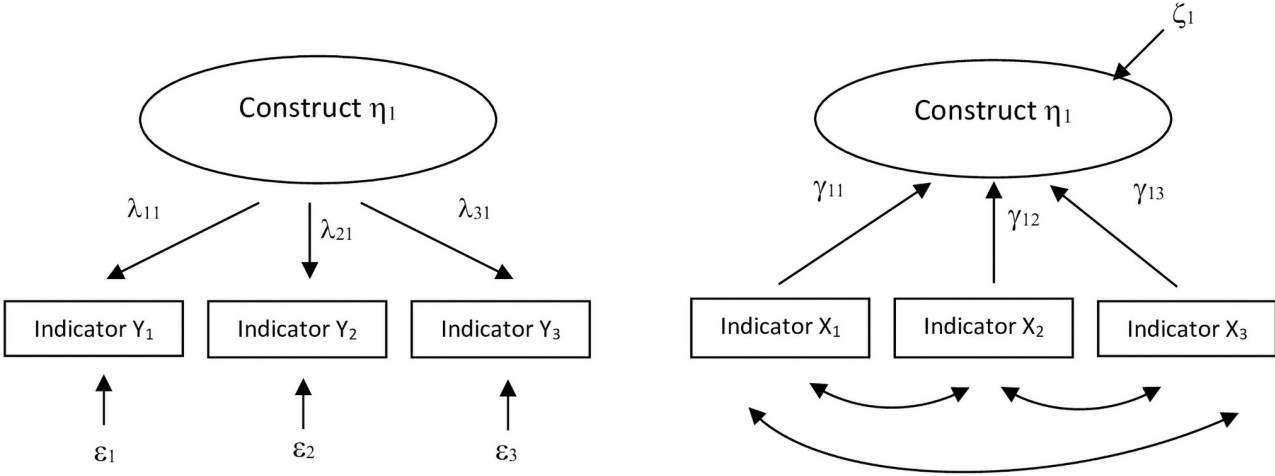

**Fig 1. Path diagrams of reflective and formative models.** A. Reflective model. B. Formative model.

driven by the latent *IM* factor. Note that the nature of the factor represents another (more profound) level than its indicators. For this reason, removing a specific reflective indicator does not alter the conceptual domain of the construct, nor would that involve any conceptual misspecification.

Likewise, it is vital to clarify which of the nine underlying dimensions can constitute a series of complementary indicators, indicating a *formative* model, or include some reflective items within the same dimension that are complementary with the others, rather than affined. Unlike the reflective model, the specific formative factor inhabits the same conceptual level as its indicators, as visualized in Fig 1, too. This means that each item (or some of them) constitutes a facet of the leadership dimension. Omitting an item often results in an incomplete picture of the specific dimension, leading to conceptual misspecification. A formative factor model therefore requires a census, rather than a sample, of interchangeable indicators [14, 49, 51, 53]. In contrast to the reflective model, causality flows from the observed variables to the construct. For example, *Individualized Consideration (IC)* is a construct that clearly contains two different sub-dimensions. Two items are focused on *IC*, which labels the factor, but the other two are related with another, more development-driven component. Hence, the actual wording indicates that by pairs the items share some specificities. Because the behavioral indicators are not driven by the same underlying facet of the skill, they will not necessarily covary among the four of them. This is also the case for many other of the nine MLQ dimensions.

Once these basic epistemic relationships are established, the next step is to clarify the nature of the three multidimensional second-order factors—transformational, transactional, and laissez faire styles—that reflect their relationships with the nine first-order factors. Following the taxonomy of Law et al. [39], the present study discusses whether this multidimensional construct and its dimensions constitute a *latent* (reflective second-order) or *emergent* (formative second-order) model. Mirroring the distinction between reflective and formative indicators, the *latent multidimensional* construct can operate at deeper and higher levels than its dimensions. Latent models can be described as constructs manifested through different dimensional forms. By contrast, the emergent multidimensional construct exists at the same level as its dimensions, which cover its relevant domains exhaustively. When researchers analyze multidimensional constructs under an emergent model, they typically have two options: aggregate operational variables, or profile operational variables [39].

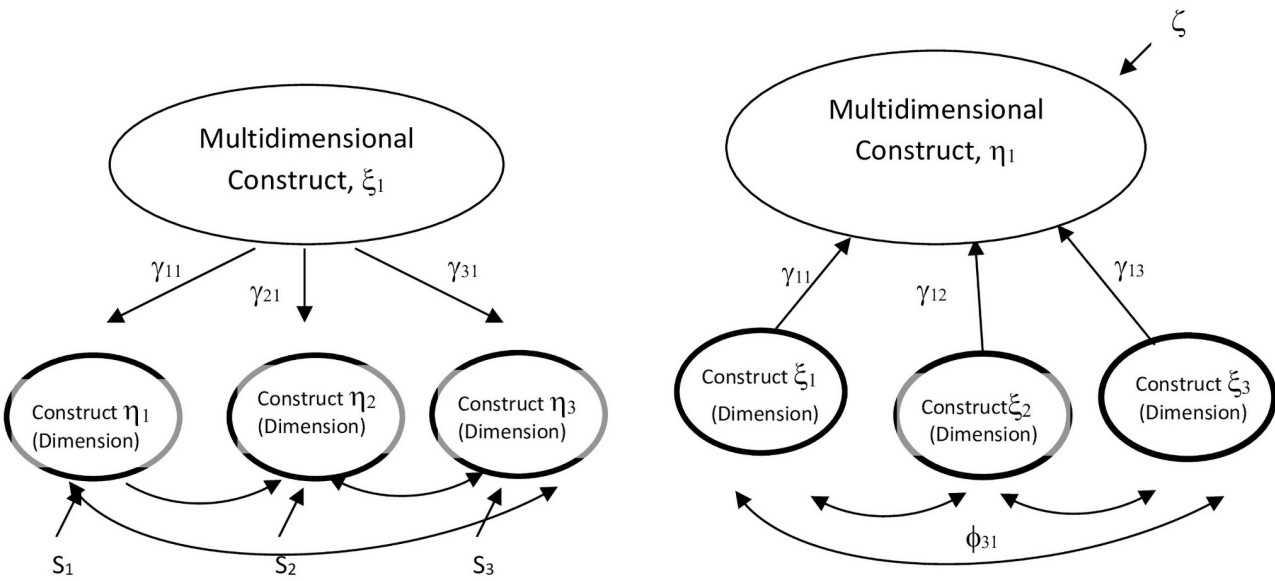

**Fig 2. Path diagrams of latent and emergent models.** A. Latent model. B. Emergent model.

Both latent and emergent aggregate multidimensional constructs can be represented by algebraic combinations of their dimensions, as can be seen in Fig 2. However, an error term is involved in the case of the aggregate construct, as with formative first-order factors. Below, we propose that the five sub-dimensions of the transformational leadership style, as well as the three sub-dimensions of the transactional style, should be understood more as aggregated multidimensional constructs than as latent. Other researchers, too, have proposed that transformational leadership must be specified as an aggregate multidimensional model [54].

Finally, it is important to clarify whether the three styles/scales that constitute the "full range of leadership" construct should be operationalized in either of the following two ways:

1. On the one hand, the construct may involve a single holistic concept, operationalized by means of a third-order factor. In this case, we must determine whether the three styles can be combined into a latent or aggregated multidimensional construct, providing a single value for the assumed third-order factor; or

2. Alternatively, on the other hand, this multidimensional construct cannot be expressed as a combination of three scales, by offering a single overall value for the multidimensional construct. In this case, the various levels of the three scales should be described as full-range leadership–that is, a profile multidimensional construct [14, 39]. As we will argue below, we think that this profile is the multidimensional construct that adapts better to the MLQ second-order factor structure–the styles.

## Method

By way of steppingstone, we have first performed a qualitative content analysis of the wording of the four items attached to each of the nine MLQ factors. Subsequently, we developed a quantitative approach to empirically assess the nature of the nine first-order factors, three second-order factors, and one global MLQ-5X factor. We tested the unidimensionality of each of the nine MLQ factors using a CFA maximum-likelihood estimation, complemented by computing the power of these tests. We would like to emphasize that researchers have taken for

granted the MLQ dimensions' unidimensionality or have omitted to assess the power of the test in using MLQ data. This could have been an issue in the previous studies verifying and interpreting of MLQ as a measurement leadership instrument.

We gathered MLQ data from two very different samples. Our first sample includes 129 police officers from the Catalan Police workforce, which were randomly chosen within each sector and territory of the police force. The average age was 44.48 (SD = 5.35), and 15.5% were female. The participants' statements about their respective leaders were used to gather MLQ-5X data on two occasions, before and after a training program led by one of the researchers of this study. Since the study was designed to highlight a gap in the MLQ instrument, rather than the effects of training, only post-program data were considered in the current paper. We decided to focus on the post-program survey because by that time participants were already familiar with the test, leading to fewer measurement errors.

The second sample comes from a survey using the online platform Prolific, which is an online research platform through which people participate in posted data collection initiatives, receiving payments in exchange. Online platforms such as Prolific allow for the recruitment of a large and diverse set of participants at a low cost, and simplify the administration of online surveys [55]. Although this recruitment does not involve a random sampling process, resulting in an issue with representative sampling, several studies have reported that findings from online research platforms are as valid and reliable as those based on traditional sampling methods [56, 57] In comparison with several other research platforms, Prolific has been found to be superior in terms of data quality, offering an option to recruit a representative sample in terms of age, gender, and ethnicity/race [58, 59]. We limited Prolific participants to representative US adults aged 18 and older.

## Analysis

### Question 1: Is MLQ a formative or a reflective model?

**MLQ content analysis.** Our first question involves the formative or reflective nature of the MLQ items. The first step in our journey is a preliminary MLQ content analysis. The MLQ is composed of nine factors. Before carrying out any formal analyses, we carefully read all the items as they were designed to associate with these nine factors. Whilst doing so, we sought to answer the following basic questions: (1) Should the four indicators within each of the nine dimensions correlate so that the usual factor analysis model is the proper specification? (2) Do we consider the indicators of each of the specific nine dimensions to be interchangeable? and (3) Would we be comfortable computing reliability by assessing the internal consistency among all items in any factor? This process does give a first hint as to whether or not the indicators can be viewed as reflective (effect) indicators. Not doing so thoroughly may lead to flawed conclusions about constructs [14, 49, 53].

The first five dimensions include elements of the then-new transformational leadership perspective proposed by Bass [25], while the next three dimensions display a transactional leadership dimensions cluster, reflecting the old leadership paradigm of the 1950s and 1960s, which focuses on tasks or people. Specifically, the first five dimensions–idealized influence attributed, idealized influence behaviors, inspirational motivation, intellectual stimulation, and individualized consideration–measure transformational leadership. The next three–contingent rewards, active management-by-exception, and passive management-by-exception–capture transactional leadership. The final factor indicates a lack of leadership (or laissez faire leadership).

A first step is to assess all assumed factors' face validity and formative/reflective nature. Regrettably, we cannot show the exact wording of the MLQ items, due to copyright issues, but

below we illustrate our interpretation of this critical content analysis for the MLQ's three leadership styles, one by one. Whilst doing so, we already refer to illustrative outcomes of our quantitative psychometric analysis, which comes next, that provide further support for our qualitative findings.

*Transformational leadership.* The first issue to consider is whether the underlying dimension of *Attributional Idealized Influence* (*AII*) is a reflective factor analysis model, manifested through a series of indicators, as is usually taken for granted. If so, the four listed items represent a sample of potentially interchangeable indicators. Alternatively, the model could be formative. In that case, two or more constitutive facets would establish the dimension. Then, the items should be the census of indicators needed to describe the *AII* construct. *AII*-related behaviors reflect leadership attributions made by followers on the basis of their own perceptions of the leader. The listed attributions range from feelings of pride to feelings of confidence, and from perceiving the leader's generosity to respecting her or him. Two items are linked through the idea of respect and admiration, another relates to group identity, and the last one involves safety or trust. The latter concepts are not necessarily linked to admiration, respect, or being a role model. Although these concepts involve the leader's impact, as perceived by subordinates, they are likely to be enhanced by a "charismatic" halo effect. As a consequence, the item loadings are likely to be somewhat inflated. In the next section, more quantitative perspective will show that a single factor model could account for the inter-correlations among the four items in a sample from a hierarchical organization such as the Catalan Police. However, in our general US population sample, although global fit indices do not reject the single-factor solution, we observe that the reduced magnitude of the last item loading would correspond more to another facet.

The *Behavioral Idealized Influence (BII)* dimension has implicitly been presented as a reflective factor. However, again, the wording of the items reflects a certain degree of diversity. Whilst the first item refers to values, the other involves the leader emphasizing the importance of the purpose, mission and ethical consequences of the decisions. In addition, the word "importance" that appears in two items is an artifact that is likely to enhance a participant's perception of consistency. This is another example of a formative factor with two sub-dimensions, which may be referred to as trust and purpose. Note that a leader could demonstrate these complementary behaviors fully or partially. In the next section, we will see that both samples show a similar loading pattern that corroborates this bidimensional structure.

The construct of *Intellectual Stimulation* (*IS*) should also be operationalized through a formative factor model, covering a range of different domains or facets. On the one hand, it provides a critical perspective on the appropriateness of assumptions; on the other hand, it captures the relevance of providing new and diverse problem-solving perspectives. In the next section, we will see that this bidimensional structure does emerge clearly in the US sample, whereas the Catalan Police sample generates a one-dimensional pattern.

Analogous to *IIA*, the factor of *Inspirational Motivation* (*IM*) refers to emotional competencies that inspire, offering a bright perspective on the future. While three items are linked with the notion of a compelling future vision that must be accomplished, one item reflects an optimistic perception of history, which has been linked to the hope construct, based on its agentic component [60]. The literature on hope and optimism argues that these two constructs are related, but are not the same [61]. Arguably, most inter-item correlations are affected by a "self-fulfilling prophecy" halo effect. Item loadings could therefore be inflated. In the next section, we will see that, although in both samples unidimensionality of these four items is not rejected, the loading of the history-focused item in both samples is clearly always the lowest.

This concludes our face content validity analysis of the transformational leadership dimensions of the MLQ, since we have already mentioned above that the items' wording of the

*Individualized Consideration (IC)* actually illustrated a two-factor model with two formative subdimensions.

*Transactional leadership*. The three transactional leadership dimensions are usually assumed to be unidimensional and reflective. However, the definition of the *Contingent Reward*s (*CR*) factor includes economic and emotional exchanges, clarifying responsibilities, and rewarding desired outcomes, which would clearly lead to a formative factor model. Three items reflect the notion of exchange rewards (aka Vroom's [62] Expectancy Theory of Motivation), while one item belongs to another dimension, linked to an individual's commitment to pursuing her or his own performance. The wording of these items is closer to the transformational leadership cluster than to the other two *Management-By-Exception* (*MBEA* and *MBEP*) factors from this transactional leadership cluster.

While all items attached to the *MBEA* dimension focus on mistakes and deviations from standards, only one item is related with the notion of actively managing a situation. In the next section, we will see that this is perceived by respondents in both samples as a different facet. Additionally, when referring to the most negative dimension of leadership, *MBEP*, respondents are likely to be influenced by the common ending statement "before taking action" in two of the four items. As we will see in the next section, this specific component inflates their correlation, which leads to the emergence of two *MBEP* subdimensions in both samples. Moreover, regarding the Catalan Police subordinates' perception of the inactivity of their commanders, 80% of the answers concentrate in the first two (0, 1) answer categories (i.e., 0 and 1). In contrast, in the US general population sample, these categories vary for all four items from 40 to 60%. This reduced variability (especially among the Catalan Police sample) would certainly affect the identification of relationships.

*Laissez-faire leadership*. Finally, *laissez-faire* is the avoidance or absence of leadership. In theory, this is the most ineffective leadership style, and the most inactive one. As we will see in the next section, although the wording reflects that three items are related to the notion of ignoring responsibilities as a leader, and one item to the notion of delaying actions, this is not perceived by the respondents as such in either sample. Moreover, in the Catalan Police sample, again, the use of a common word ("avoid") in two items artificially produces shared specificities in the factor model due to more consistency across the answers from the subordinates. A final remark relates to the observation that to the two *MBE* factors in the transactional leadership part of the MLQ that have negative and coercive connotations brings them closer to the non-leadership or *laissez faire* style cluster than to the *CR* factor within the same transactional leadership cluster. Actually, in the last version of MLQ, *MBEP* is clustered together with the laissez faire style.

**MLQ psychometric analysis.** *Baseline analysis*. The above content analysis of MLQ's face validity suggests that the assumed MLQ factor structure is probably not justified. Actually, the answers to the sets of four items attached to each dimension are not necessarily consistently correlated, based on close reading of the wording of the items. Instead, the content analysis above would certainly suggest that we have to customize the specification of this formative nature of the MLQ first-order factors, rather than to stick to specifying the usually taken-for-granted set of reflective factors. Hence, in this section, we assess the psychometric properties of the MLQ for the two different samples.

Interestingly, Table 1 shows that, had we conducted the standard test of the unidimensionality of the four items attached to each MLQ factor, only a few of them would have been rejected, implying that many earlier studies wrongly continued on the basis of the (implicit) assumption that each of the nine dimensions of MLQ are properly specified as a single-factor analysis model, with four reflective items for each dimension. Notice that the 0–4 (or 1–5) answer modality used in MLQ actually implies ordinal properties of the data gathered.

**Table 1. Comparison of the global fit indices and sensitivity test of the nine MLQ dimensions.**

**A. Catalan Police (n = 130)**

| Dimension | SB-$\chi^2$(2) | RMSEA | P$_{Close}$ | CFI | SRMSR | Loadings | (Power) | EPC |
|---|---|---|---|---|---|---|---|---|
| IIA | 0.66 | .00 | .79 | 1.00 | .014 | .68;.51;.82;.63 | (0.14) | -.08 |
| IIB | 1.41 | .00 | .36 | .990 | .031 | .32;.83;.68;.71 | (0.01) | -.27 |
| IS | 0.08 | .00 | .97 | .985 | .031 | .26;.46;.62;.73 | (0.02) | -.12 |
| IM | 0.53 | .00 | .52 | 1.00 | .021 | .50;.76;.61;.82 | (0.14) | -.08 |
| IC | 9.22 | .17 | .09 | 1.00 | .049 | .72;.36;.37;.65 | (0.04) | -.34 |
| CR | 3.52 | .08 | .28 | .967 | .033 | .28;.38;.66;.57 | (0.11) | -.22 |
| MBEA | 6.49 | .13 | .09 | .952 | .053 | .56;.80;.65;.29 | (0.07) | -.54 |
| MBEP | 6.02 | .12 | .11 | .890 | .056 | .19;.63;.43;.56 | (0.02) | -.78 |
| LF | 7.61 | .14 | .05 | .987 | .053 | .85;.35;.80;.39 | (0.11) | -.37 |
| | (a) | (b) | (c) | (d) | (e) | (f) | (e) | (g) |

**B. US population (n = 300)**

| Dimension | SB-$\chi^2$(2) | RMSEA | P$_{Close}$ | CFI | SRMSR | Loadings | (Power) | EPC |
|---|---|---|---|---|---|---|---|---|
| IIA | 1.75 | .001 | .64 | .997 | .013 | .83;.83;.87;.41 | (0.26) | -.16 |
| IIB | 3.13 | .043 | .43 | .997 | .020 | .52;.84;.64;.80 | (0.18) | -37 |
| IS | 2.38 | .025 | .54 | 1.00 | .015 | .66;.72;.83;.76 | (0.28) | -.21 |
| IM | 6.20 | .084 | .17 | .990 | .023 | .66;.79;.85;.75 | (0.35) | -.36 |
| IC | 9.84 | .110 | .05 | .980 | .034 | .75;.68;.47;.85 | (0.11) | .81 |
| CR | 14.3 | .140 | .01 | .967 | .049 | .69;.59;.67;.77 | (0.18) | .29 |
| MBEA | 5.40 | .075 | .22 | .989 | .053 | .73;.42;.71;.72 | (0.12) | .33 |
| MBEP | 7.76 | .098 | .10 | .984 | .033 | .66;.82;.42;.78 | (0.02) | -.78 |
| LF | 3.66 | .084 | .36 | .995 | .022 | .74;.66;.74;.80 | (0.14) | -.21 |
| | (a) | (b) | (c) | (d) | (e) | (f) | (e) | (g) |

Two-factor model (constraining PS(1,2); $\psi_{21}$ =1), (a) SB-Scaled Chi-square (degrees of freedom); for the single-factor model, (b) RMSEA, (c) probability of close fit for RMSEA (d) Comparative Fit Index; (e) SRMSR = standardized root mean square residual; (f) loadings of the four items; and (g) (power of the test) and EPC = estimated value for misspecification $\psi_{21}$ =1 and (power of the test).

For Polychoric, no change in the fit of most of the subdimensions; just loadings are slightly higher. The only significant differences are: a much better fit for *LF* SB-Scaled Chi-Square = 1.79 (p = 0.41) & p-value for test of close fit (RMSEA < 0.05) = 0.52; for *IC* SB-Scaled Chi-Square = 0.79 (p = 0.67) & p-value for test of close fit (RMSEA < 0.05) = 0.75; for *MBEA* SB-Scaled Chi-Square = 0.70 (p = 0.71) & p-value for test of close fit (RMSEA < 0.05) = 0.77; and *MBEP* SB-Scaled Chi-Square = 2.03 (p = 0.36) & p-value for test of close fit (RMSEA < 0.05) = 0.47.

Two-factor model (constraining PS(1,2); $\psi_{21}$ =1)), (a) SB-Scaled Chi-square (degrees of freedom); for the single-factor model, (b) RMSEA, (c) confidence interval, and (d) probability of close fit for RMSEA; (e) SRMSR = standardized root mean square residual; (f) loadings of the four items; and (g) (power of the test) and EPC = estimated value for misspecification $\psi_{21}$ =1 and (power of the test).

For Polychoric, no change in the fit of the subdimensions; just loadings are slightly higher.

Therefore, the Polychoric correlation matrix should be the one to be analyzed. However, since the standard practice is to use the Pearson covariance correlation matrix for the analysis, we provide the results of this second approach, and make explicit comments when we observed relevant differences between both analyses. In general, we could say that the appropriate Polychoric correlations lead to a better fit of the CFA model, as well as higher loading estimates. However, differences in the US population sample are lower than in the Catalan Police sample. In the latter, the model actually lost even more sensitivity to detect unidimensionality misspecifications. Indeed, in both samples, using Maximum Likelihood estimation on the covariance matrix (LISREL8.80), the usual global fit indices (all above .90, and most of them above .95), apart from the RMSEA 95% Confidence Interval, would only reject the single-factor solution for *IC* among the set of five transformational leadership dimensions. Actually, in both samples,

a single-factor solution leads to bipolar patterns of loadings: the *IIA* exhibits three loadings of high magnitude and one very low; two of the *IIB* items' loadings are high, and the other two are very low; the *IS* loadings pattern differs between samples, but both displays a two-factor solution–two items per factor in the Catalan Police sample, and one item being heterogeneous in the US citizen sample; finally, *IM* loadings show that one item is not consistent with the other three.

Regarding the three transactional leadership dimensions, our two samples give different results for the *CR* dimension (see Table 1(a) and 1(b)). Unidimensionality is not rejected in the Catalan Police sample, but clearly is so in the US population sample. Strikingly, this result is in sharp contrast with the pattern of loadings in both samples. On the one hand, the Catalan Police sample shows a pattern that matches a bifactorial structure. On the other hand, the US population sample might be derived from a single-factor solution. Also, *MBEA* reveals an opposite pattern of loadings. While the second item "Keeps track of all mistakes" has the highest loading for the Catalan Police sample, it is the lowest in the US citizen sample. Clearly, *MBEP* is associated with a bidimensional pattern in both samples. Finally, *LS*, again, comes with an opposite result in both samples. In the general US population sample, a single-factor solution is not rejected, while the Catalan Police subordinates clearly perceive two different dimensions. Arguably, as discussed in the previous section, we believe that this is due to the influence in the police subordinates' perception of their leader of the word "avoid".

Some differences between samples regarding their model fit and the item loading values have plausible explanations, following from the different characteristics of both samples. However, we cannot think of any plausible explanation for the non-rejection of unidimensionality in either or both samples of those dimensions that are, according to both content and psychometric analyses, clearly bidimensional. Our argument is that the implicit assumption that MLQ is a reflective model is to blame. But before exploring this argument, we first turn to assess the model sensitivity in a more detailed analyses of the tests' power in different parts of the model.

*Model sensitivity*. Saris, Satorra, and van der Veld [63] have shown that, when the suggestions proposed by Hu and Bentler [64] are adopted, the behavior of many fit indices is very poor. In particular, $\chi^2$ and other fit indices are associated with different degrees of sensitivity for different misspecifications of the model. Model rejection can be caused by minor misspecifications when the test is very sensitive (high power); conversely, very large misspecifications can fail to trigger rejection when the test is insensitive (low power). For this reason, the usual testing procedure must be combined with a sensitivity analysis of the test statistics for all possible misspecifications [63]. Hence, we have assessed model sensitivity of these global tests for this specific misspecification because (a) the textual analysis, the correlation pattern among the four items in each dimension as well as their loadings do clearly suggest that the single-factor solution is unlikely to work, and (b) of the inability of the global fit tests (Table 1) to detect this misspecification in our data, but also in the literature.

To compute sensitivity, we first specified the hypothesized "correct" solution, which is a two-factor structure for each subdimension. Next, within each subdimension, we restricted the correlation between the two factors to 1 –i.e., $\psi_{21}$ =1. This actually restraints the two factors to collapse into the assumed single latent factor model underlying the nine MLQ subdimensions. The global test results for both samples reported in Table 1(a) and 1(b) show that the global fit indices are generally insensitive in many of the nine MLQ dimensions to this particular misspecification that imposes $\psi_{21}$ =1. Indeed, column (g) reveals the test's low sensitivity for this misspecification. The reduced power values in column (g) show why the global fit indices are insensitive and, consequently, why they fail to reject the single-factor model. This surely

**Table 2. Correlations among the nine MLQ subdimensions.**

A. Catalan Police (n = 130)

| | IIA | IIC | IM | IS | IC | CR | MBEA | MBEP | LF |
|---|---|---|---|---|---|---|---|---|---|
| IIA | 1.000 | | | | | | | | |
| IIB | **.49/.74** | 1.000 | | | | | | | |
| IM | **.63/.82** | **.63/.74** | 1.000 | | | | | | |
| IS | **.43/.75** | **.39/.68** | **.41/.76** | 1.000 | | | | | |
| IC | **.58/.82** | **.48/.74** | **.55/.82** | **.46/.75** | 1.000 | | | | |
| CR | .65/.91 | .60/.83 | .67/.92 | .47/.84 | .56/.92 | 1.000 | | | |
| MBEA | -.05/.03 | -.02/.03 | -.14/.03 | .03/.03 | -.03/.03 | **.07/.03** | 1.000 | | |
| MBEP | -.35/-.43 | -.23/-.39 | -.29/-.44 | -.07/-.39 | -.22/-.43 | **-.20/-.46** | **.09/-.01** | 1.000 | |
| LF | -.53/-.43 | -.34/-.50 | -.38/-.55 | -.20/-.50 | -.23/-.54 | -.38/-.68 | .09/-.02 | **.52/.77** | 1.000 |

B. US population (n = 300)

| | IIA | IIC | IM | IS | IC | CR | MBEA | MBEP | LF |
|---|---|---|---|---|---|---|---|---|---|
| IIA | 1.000 | | | | | | | | |
| IIB | **.72/.76** | 1.000 | | | | | | | |
| IM | **.76/.79** | **.73/.82** | 1.000 | | | | | | |
| IS | **.75/.77** | **.72/.75** | **.73/.74** | 1.000 | | | | | |
| IC | **.84/.80** | **.74/.73** | **.75/.74** | **.82/.81** | 1.000 | | | | |
| CR | .80/.78 | .71/.70 | .74/.74 | .73/.74 | .80/.76 | 1.000 | | | |
| MBEA | -.19/-.09 | -.02/-.04 | -.07/-.05 | -.02/-.03 | -.12/-.08 | **-.04/.05** | 1.000 | | |
| MBEP | -.62/-.55 | -.54/-.48 | -.52/-.47 | -.50/-.44 | -.59/-.51 | **-.57/-.52** | **.13/.06** | 1.000 | |
| LF | -.66/-.66 | -.54/-.53 | -.53/-.55 | -.49/-.51 | -.58/-.59 | -.60/-.59 | 17/.14 | **.78/.75** | 1.000 |

Notes. Computed as parcels or as a /CFA model factor. Numbers in bold refer to intra-style sub-dimensions.

Computed as parcels or as /CFA model factor

explains why earlier work that makes frequent use of the MLQ has failed to notice the key mis-specification in the underlying factor analysis model pointed out in this paper.

Were this restriction to be relaxed, the actual Estimated Parameter Change (EPC) value would always lead to a very low value of the correlation between the two factors ($\psi_{21}$), in some cases negligible. These results show that most of the nine MLQ factors are not reflective but formative, subsuming within each factor two or three sub-dimensions or facets. In all, these findings resolve our first question, showing that the factorial structure of the five first-order transformational leadership factors and three transactional leadership factors is not purely reflective, but to a great extent formative. The insensitivity of the usual global tests to this mis-specification could explain why other researchers failed to detect this flaw in the MLQ factorial structure.

*A formative versus reflective model.* The matrices displayed in Table 2 report the correlations among the nine MLQ dimensions, based on two different assumptions. First, we consider the nine dimensions to be formative first-order factors. Therefore, the items are clustered into parcels, and their factors are computed as summative rating scales (SRS) of equal weights. This is the usual procedure in cases where there is no substantive information about the weights in a formative model. Second, we consider the nine MLQ dimensions as latent single factors, the items being their reflective indicators. This is the model usually specified in analyses of MLQ data, leading to factor scores estimated using CFA models.

Table 2 reveals three relevant issues. First, had we specified a nine-factor reflective model, as researchers usually do, the correlations among the nine dimensions would be greater than

those achieved using formative factors, computed via SRS. These differences seem clearer in the Catalan Police sample. Second, independent of the approach chosen or the sample considered, the *CR* factor has higher correlations with the five transformational leadership dimensions than within its own-cluster transactional leadership dimensions. Actually, the correlation between *CR* and *MBEA* is negative and negligible; with *MBEP*, it is reduced, becoming negative. Third, besides this last correlation within the cluster of transactional style components, the greater magnitude and positive correlation of *MBEP* with the laissez faire style in both samples constitute further evidence of the MLQ's inconsistencies regarding the underlying MLQ factorial structure. These have been typical results in many MLQ studies.

## Question 2: Is MLQ a latent or an emergent model?

The second question involves whether the three second-order factors (transformational, transactional, and laissez faire leadership styles) lead to a multidimensional *latent* or *emergent* model. While the MLQ model has always been specified as *latent*, we argue that the alternative *emergent* specifications for multidimensional constructs are better suited to the MLQ. Instead of considering the nine basic MLQ dimensions as sample reflections of latent leadership styles, we would argue that the first-order factors (e.g., *IIA*, *IIB*, *IS*, *IC* and *MI*) are separate facets that characterize an individual's transformational leadership style. Each style is therefore represented by an *emergent aggregate* multidimensional construct at the same level of abstraction as its constitutive dimensions. This is even clearer with the sub-dimensions of the transactional leadership style, as their items' wording and negligible correlations reveal that they refer to different, incompatible, and even antagonistic behaviors.

In the management literature, aggregate multidimensional constructs are not unusual. For example, MARKOR [65] defines market orientation as a composite of dimensions, including intelligence generation, intelligence dissemination, and responsiveness. Locke [66] develop job satisfaction as a composite made by aggregating satisfaction with four different work factors. Also, Hackman and Oldham [67] propose a motivating potential index that aggregates five job characteristics.

There are many differences between what we suggest here, which views the MLQ as an *aggregate* model, and the usual implicit assumption that the MLQ captures a multidimensional *latent* construct. All the above clearly supports the former and undermines the latter. Disregarding one of these basic dimensions in an *aggregate* model changes the conceptualization of the leadership style in question, leading to a conceptual misspecification. In addition, changing one dimension does not necessarily change other dimensions of the *aggregate* multidimensional construct. It can therefore be assumed that the dimensions nested within an *aggregate* multidimensional construct do not need to covary [43]. Moreover, unlike the latent construct, in which basic dimensions constitute a sample, the *aggregate* multidimensional construct requires an exhaustive definition and measurement, with no dimension missing. From the perspective of this *aggregate* multidimensional framework, leadership style is operationalized as an algebraic composite of sub-dimensions plus an error term.

The relative weight of each sub-dimension in the multidimensional *aggregate* construct is independent of the covariance structure, unlike in the case of a latent construct. This implies that they must be estimated, except in the exceptional case where the exact algebraic function of the multidimensional construct (the relative weight of the basic dimensions) can be theoretically determined [39]. As in other cases (e.g. [14]), we do not consider the current theoretical development to be detailed enough to prescribe such an exact algebraic function. Taken in isolation, aggregate models such as those shown in Fig 2 are statistically under-identified—there is insufficient information to estimate every parameter in the model. This reflects

indeterminacies associated with the scale of measurement and the disturbance term of the multidimensional construct. However, alternative options for parameter estimation using Structural Equation Modeling (SEM) can be found in the literature [53]. As Table 3 indicates, estimates among the three multidimensional constructs or styles and their basic components are very different when the styles are considered to be a *latent* model (as they typically are) or *emergent* model–or *aggregated* or *profile* model–as we propose. The former approach leads to higher loadings.

As discussed above, due to lack of sensitivity for some misspecifications and indeterminacies in the measurement model, meaningfully testing relationships among constructs is impossible [49]. It is therefore not surprising that none of the standard global fit indices in SEM rejected either model in the smaller sample. The global goodness-of-fit indices are actually very good, even when both approaches–SRS and CFA–provide an estimate of the correlation between the transformational and transactional second-order factors that is greater than 1, which is clearly an improper solution and a sign of empirical under-identification. In accordance with Saris et al.'s [63] proposition, we have tried to detect misspecification errors, rather than focusing solely on global fit. As a result, and independent of the specification used (the CFA *latent* model or the CFA *aggregate* model based on parcels), both analyses detected two clear misspecifications inherent to the MLQ model.

The first misspecification relates to *CR*, considered a factor of the *latent* transactional leadership style. This is identified in our data as a model misspecification by being much more associated with the *transformational* leadership multidimensional construct (see the between and within correlations among the nine factors assessed by both samples in Table 2). Specifying *CR* as a transactional leadership component leads to an improper correlation between the second-order transformational and transactional leadership factors greater than 1. Irrespective of whether factor scores or SRS are used for computing factors scores, either using Pearson or Polychoric correlations, the improper solution persists. However, our relatively small Catalan Police sample (n = 130) is responsible for this improper solution, because the model is relatively complex. In the bigger US general population sample, both factor specifications and both analyzed covariance matrices lead to an identical correlation (0.99) between these second-order factors.

This raises the question as to whether and why the same improper solution would also result from the parsimonious *aggregate* model. We would argue that misspecifications are major sources of improper solutions [68]. These improper solutions are likely due to low factor loadings (in our case, poor transactional leadership item loadings and scale reliability), which magnify the effect of the higher inter-correlation of *CR* with the transformational (rather than transactional) leadership components.

If a constraint is added in our Catalan Police sample to restrict the estimated correlation between the transactional and transformational leadership factors to be below 1 [69], we produce a non-convergent solution, probably because of under-identification (see [68, 70]). This is analogous to the well-established issue with ipsative data [71] where the forced-ranking effect gives correlations (negative or higher than 1) among the three second-order factors that are artifacts of the improper method of measurement. The second misspecification in both samples reflects the fact that *MBEP* has a higher correlation with *LS* than with *CR* or *MBEA*. This finding implies that we should not treat MLQ styles as *latent* multidimensional models. Our results in Table 4 show strong support for treating the three second-order factors (transformational, transactional, and laissez faire leadership styles) as multidimensional emergent models, which have five, three (or two, if we consider that *MBEP* rather associates with the laissez faire style), and one (or two) component(s), respectively (As we will discuss in the Conclusion, in the last version of the MLQ, we found that *MBEP* is captured along the laissez faire

**Table 3. Global fit indices for the MLQ second-order factor model.**

| Sample/Factor model | SB-$\chi^2$(24) | RMSEA | P$_{Close}$ | CFI | SRMSR | Factor correlations 1–2; 1–3; 2–3 |
|---|---|---|---|---|---|---|
| US300/Factor | 132.0 | .123 | .000 | .977 | .040 | .99;-.65;-.68 |
| US300/SRS | 94.61 | .099 | .000 | .988 | .035 | .96;-.64;-.66 |
| Police130/SRS | 27.69 | .035 | .647 | .996 | .048 | .99;-.55;-.63 |
| Police130/Factor | 32.98 | .054 | .407 | .987 | .054 | .99;-.51;-.35 |
| | (a) | (b) | (c) | (d) | (e) | (f) |

MLQ three second-order factors' model using CFA or SRS for factor scores computation: (a) Satorra-Bentler Chi-square (degrees of freedom), (b) RMSEA, (c) probability of close fit for RMSEA, (d) Comparative Fit Index, (e) standardized root mean square residual, and (f) correlations among the MLQ three second-order factors.

leadership style.). In these cases, the components of each style are complementary; there is no reason for them to be correlated. Assigning this very different specification to relationships between the three main constructs and their dimensions has a profound effect on theory development and testing, involving the relationships between other constructs and the multidimensional construct of interest [15, 72].

## Question 3: Is MLQ a profile model?

Finally, this leads to the third question, which focuses on the multidimensional structure of the three MLQ leadership styles. It is clear that a *latent* multidimensional model cannot represent this type of relationship, as the inter-correlations between the dimensions of the three styles observed in both samples included in Table 2 are negligible. So, if the three leadership styles operationalized through the MLQ do not constitute a *latent* multidimensional construct, what multidimensional construct model best represents the assumed full-range leadership theory? On the one hand, we have explained why the MLQ multidimensional construct has been traditionally consider to be *latent*. On the other hand, the current theoretical development of the MLQ does not suggest any exact algebraic function as a multidimensional emergent aggregate construct. Is there any other plausible alternative?

In providing a full-range leadership theory, or a transformational-transactional-laissez faire leadership theory, Avolio and Bass [73] built a multidimensional construct that includes antagonistic leadership behaviors. Studies usually present these three leadership styles as a sequential and progressive gradation, from laissez faire via transactional to transformational. However, instead assuming that leadership is a contingency-driven learning process, we recommend a

**Table 4. Comparison of the loadings of the sub-dimensions on the three second-order factors as latent or as aggregate.**

| A. Catalan Police (n = 130) | |
|---|---|
| **Dimension** | MLQ nine first-order factor loadings |
| | Transformational//Transactional//Laissez faire |
| Latent | .90.81.91.83.90//.98.03 -.47//1. |
| Aggregate | .79.71.82.54.69//.63 -.10 -.25//1. |
| **B. US population (n = 300)** | |
| **Dimension** | MLQ nine first-order factor loadings |
| | Transformational//Transactional//Laissez faire |
| Latent | .95.94.87.89.96//.92 -.17 -.64//1. |
| Aggregate | .91.85.87.86.87//.88 -.01 -.57//1. |

portfolio of styles, rather than viewing the transformational style as the progressive end point. Then, we can consider a plausible alternative to the pattern of expected relationships in the comprehensive full-range theory model: We argue that the three leadership styles identified by theory and measured by the MLQ actually correspond to a *profile* multidimensional construct in line with the taxonomy developed by Law et al. [39]. We further discuss this suggestion in the Conclusion.

## Conclusion

A proper conceptual specification of any measurement model is critical. The MLQ is associated with measurement model misspecification, which is a source of both biased structural parameter estimates and poor fit in covariance structure models [41]. The ritualistic specification of the nine MLQ first-order factors as factor analysis *reflective* models instead of specifying them as *formative* involves such a major misspecification. Both theoretical and empirical considerations suggest that the first-order formative factor model is more plausible than the reflective factor model—and that the second-order *aggregated multidimensiona*l model is more plausible than the *latent multidimensional* alternative. We therefore suggest that future studies should consider conceptualizing the nine first-order dimensions of the MLQ as a mix of reflective and formative sub-dimensions, with the three second-order factors as *aggregated multidimensional* models. Finally, we argue that full-range theory should be specified as a *multidimensional profile* model, rather than as the usual gradation (or progression) of leadership styles.

Generally, theory-based empirical research on leadership and full-range theory focused on estimating and testing the relationships between dimensions of the theory, other constructs of interest (e.g., efficacy or satisfaction) within the same MLQ, and other leadership constructs, including participative leadership [74], personality and performance [75], empowering leadership [76], ethical leadership [77], LMX [78], and perceived supervisor support [79]. Few studies have explored the conceptual specification of the full-range-theory's multidimensional constructs. In his critical review of TCL history, Antonakis [10] described this theory as a necessary breakthrough in "a time where there was pessimism and no direction in leadership research . . . it provided leadership researchers the 'ah-ha' moment for which they had been waiting for many years" (p. 257). Van Knippenberg and Sitkin [12] have critiqued its conceptualization, noting that "there does not seem to be a conceptually sound and bounded definition of charismatic-transformational leadership" (p. 4). In the same vein, Antonakis et al. [80] have outlined conceptual problems using existing definitions. These measurement misspecifications have led to erroneous parameter estimates and misleading statistical tests.

Consequently, the main purpose of our study was to assess the risks of conceptual misspecification in constructs and to propose guidelines for improving the conceptual specification of full-range theory constructs in future research. The present study also helps to explain why these problems have not been detected empirically, despite many criticisms of this issue in the literature (e.g., [11, 13]). Some of the stated inconsistencies can be attributed to the assumed factor structure of the MLQ-5X. An adequate specification of the MLQ's underlying model would contribute to the so-called "scientific success" of the full-range theory [9]. Our study reveals that the MLQ should be classified as formative, and hence not be assessed based on the degree of internal consistency. The traditional internal consistency-based statistical methods used to analyze interdependence, such as EFA/CFA and Cronbach's alpha, are meaningless and inappropriate here, as are tests of convergent or discriminant validity and reliability estimates [14, 45, 49, 53, 81, 82]. The bias of the parameter estimates can be either positive or negative [51].

Latent and aggregated models, despite their opposing structural directions (appearing in path diagrams as arrows pointing to and away from the dimensions), can both be analyzed using a covariance structure analysis. Although the MLQ has been tacitly assumed to be a latent model, Law and Wong [72] have demonstrated that different conclusions may be reached when the relationship between the construct and its dimensions is defined differently. In other words, depending on the specification chosen (latent or aggregated model), estimates of structural parameters will be diametrically different for the two models [15]. Our results only partially corroborate these conclusions. On the one hand, the relationship between the nine components and each of the three second-order factors differed because there were correlations among the factors, being estimates from the latent model and therefore greater than the proposed aggregated one. On the other hand, SEM global-fit indices were unable to identify the right model, as both fitted the data equally well. So, they were insensitive to the different misspecifications of the three multidimensional constructs.

Generally, our study illustrates the dynamic nature of measurement instruments, here revealed for the case of the MLQ. Once introduced, an instrument should not be taken for granted, but needs to be adapted, changed, and finetuned on the basis of emerging evidence, without taking implicit assumptions for granted. To produce such evidence, any instrument should be subject to systematic psychometric scrutiny in the form of series of studies re-examining the instrument's model, reliability, and validity. Indeed, in line with this, the last version of the MLQ has responded to merging critique by including *MBEP* in the laissez faire leadership style. The present study is an example of the type of work needed. But of course, as any study, ours has several potential limitations, of which we would like to briefly discuss two.

The first involves the validity of the statistical conclusions. The relatively small sample sizes and the low reliability of some of the MLQ measures imply the possibility of a low statistical power threat. This can result in a type II error: Not being able to detect effects that are actually present. The low reliability of some MLQ items, because of the five-point answer modality and the non-unidimensionality of most of the nine factors, is compensated, to a certain extent, by the common method variance effect [83]. This is why we have been so thorough in our discussion of the power of the test, rather than only focusing on the statistical significance of the results. Moreover, our results show clear evidence of construct validity. The findings agree with the sign and magnitude of expected correlations among items and first-order factors, depending on the nature of those factors (whether reflective/formative or latent/emergent).

The second is that in one of our studies we analyze (Catalan) police force data, hence collected within a very hierarchical organization. In this context, the leader has more opportunities to deploy certain behaviors and the subordinates have more opportunities to observe them than would usually be the case in other organizations, where the leader may be less exposed and her/his influence may be less acute. For this reason, the present conclusions may not be generalizable to contexts with different organizational conditions. This is precisely what observe in the general US population sample. Although the pattern of findings regarding the factorial structure of MLQ is similar across both samples, we also have detected that US general population respondents perceived the nine dimensions as more related than their police force sample counterparts. This is likely to be due to a stronger halo effect among the US sample respondents, for two reasons. First, the police subordinates were evaluating their "current chief", which has not necessarily always been the case in the US sample. And second, the MLQ was administered twice within police force for evaluating the effect training, so they very likely were able to better differentiate between the MLQ dimensions in the "post-test" that we used in the current study. We therefore recommend replicating our study in different organizations, cultures, and countries, to ensure that the criticisms identified in this paper can be fully generalized to the MLQ irrespective of the specifics of the sampling context.

Considering the popularity of the MLQ-5X instrument among leadership scholars, the present study could have significant consequences. It provides empirical proof that factor analyses and reliability measures should not be blindly based on the (implicit) assumption of reflective unidimensionality. Future studies should reconsider the use of this widely-known instrument, and enhance its construct validity in the following ways: (1) by considering the factorial structure of transformational and transactional leadership styles as formative, rather than reflective; (2) by specifying the multidimensional constructs' transformational, transactional, and laissez faire leadership styles using emergent-aggregated models; and (3) by exploring the multidimensional MLQ full-range construct through an emergent-profile model. The present findings threaten the assumed factorial structure of the MLQ, which has dominated leadership research in recent decades. The reexamination of the MLQ factorial structure provides a clear path toward understanding the way in which leadership influences organizational behavior, highlighting the need to develop more robust studies in this important research domain.

## Supporting information

**S1 Data.**
(XLSX)

**S1 File. Catalan Police force data 1.**
(DOCX)

**S2 File. Catalan Police force data 2.**
(DOCX)

## Acknowledgments

We thank Núria Aymerich and Jordi Vilardell for providing us the Catalan Police Forces data.

This study was reviewed and approved by the Committee for the Use of Human Subjects in Research (CUHSR) at ESADE Business School, case 010/2020. Written consent was gathered for each participant before starting the survey. Participants were informed of the general purpose of the study, the expected length of the questionnaire, the research team in charge of the study, the confidentiality details on how their responses would be treated, the copyright issues with the questionnaire items, and finally the ethical approval details of the study.

## Author Contributions

**Conceptualization:** Joan Manuel Batista-Foguet, Marc Esteve, Arjen van Witteloostuijn.

**Data curation:** Joan Manuel Batista-Foguet.

**Formal analysis:** Joan Manuel Batista-Foguet.

**Funding acquisition:** Joan Manuel Batista-Foguet, Marc Esteve.

**Investigation:** Joan Manuel Batista-Foguet.

**Methodology:** Joan Manuel Batista-Foguet.

**Project administration:** Joan Manuel Batista-Foguet.

**Writing – original draft:** Joan Manuel Batista-Foguet, Marc Esteve, Arjen van Witteloostuijn.

**Writing – review & editing:** Joan Manuel Batista-Foguet, Marc Esteve, Arjen van Witteloostuijn.

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
