## [Decision Letter · Decision Letter 0]

4 Mar 2021

PONE-D-20-05815

MEASURING LEADERSHIP: An Assessment of the Multifactor Leadership Questionnaire

PLOS ONE

Dear Dr. Batista-Foguet,

Thank you for submitting your manuscript to PLOS ONE. After careful consideration, we feel that it has merit but does not fully meet PLOS ONE’s publication criteria as it currently stands. Therefore, we invite you to submit a revised version of the manuscript that addresses the points raised during the review process.

We look forward to receiving your revised manuscript.

Kind regards,

Rodrigo Ferrer, Ph.D.

Academic Editor

PLOS ONE

Journal Requirements:

2. Please provide additional details regarding participant consent. In the ethics statement in the Methods and online submission information, please ensure that you have specified (1) whether consent was informed and (2) what type you obtained (for instance, written or verbal, and if verbal, how it was documented and witnessed). If the need for consent was waived by the ethics committee, please include this information.

4.We note that the grant information you provided in the ‘Funding Information’ and ‘Financial Disclosure’ sections do not match.

5. Please amend either the title on the online submission form (via Edit Submission) or the title in the manuscript so that they are identical.

Additional Editor Comments:

Dear Authors:

First of all, allow me to apologise for the delay in responding to your paper, but it was extremely difficult to find reviewers and, moreover, it has been difficult for me to make a decision, since, on one hand, I find the approach taken very interesting and novel (discussing whether the models are formative or reflective seems to me something extremely necessary, but, despite this, it is very infrequent) and, on the other hand, I see that it has important methodological restrictions, many of which I find difficult to remedy.

I think the most interesting analysis the article has is that of content review, but without presenting the items, it is impossible to appreciate their value and make them useful beyond people who are broadly familiar with the MLQ (which would make the article relevant for a leading journal, but not for a broad audience such as PlosOne). Additionally, some of the psychometric analyses used are inadequate (or, at least, their adequacy has not been made explicit), e.g. the estimation method used is inadequate for ordinal variables, it is not specified whether the polychoric correlation matrix is used, relevant fit indicators are not reported, principal components should be used for the formative model and not CFA, among other things that can be remedied. However, the major restriction to carry out these analyses is the limited sample size, so that any structure could be the product of mere chance, and it is necessary to increase it or provide other guarantees that give greater certainty in the results.

As you will see, the task is a difficult one and I leave it to you to decide whether to go ahead with the review or to look for a journal that better suits your work. Frankly, I find your work very interesting, but I need it to be redesigned

Reviewers' comments:

Reviewer's Responses to Questions

**Comments to the Author**

1. Is the manuscript technically sound, and do the data support the conclusions?

Reviewer #1: Yes

Reviewer #2: Partly

2. Has the statistical analysis been performed appropriately and rigorously? 

Reviewer #1: Yes

Reviewer #2: Yes

3. Have the authors made all data underlying the findings in their manuscript fully available?

Reviewer #1: Yes

Reviewer #2: No

4. Is the manuscript presented in an intelligible fashion and written in standard English?

Reviewer #1: Yes

Reviewer #2: Yes

5. Review Comments to the Author

Reviewer #1: The aim of the manuscript was to investigate the factorial validity of the Multifactor Leadership Questionnaire.

The authors must be commended for conducting a very thorough and original analysis of the factorial validity of the MLQ by investigating the MLQ as a reflective and a formative model.

The introduction covers the relevant literature and frames the aims of the study very well.

The authors provide an interesting and highly relevant discussion of the face validity of the items of the MLQ that are used to operationalize subscales of the MLQ. The authors state that they cannot present the individual items of the MLQ due to copyright issues. This is a big shame, as it is quite difficult to follow the discussion of the face validity of the items. Therefore, the authors should decide whether to 1) omit the discussion of the face validity, as it makes little sense in the way it is presented in the current version of the manuscript, 2) provide a more thorough description of the contents of the items (e.g. in a table), so that the reader is actually enabled to follow the discussion or 3) to present the items and deal with the copyright issues. As it stands now, this reviewer does not find that the basis of the discussion of the face validity of the MLQ satisfactory.

Another general comment is that this reviewer is a bit uncertain about the target group of the study, which leaves me wondering, why it was submitted to a general journal as PlosOne. First, the paper deals with a very specific problem regarding the measurement of leadership behaviors, which makes this reviewer think that the paper could have been more appropriately submitted to a leaderships journal. Second, the paper has its’ focus on very specific issues regarding construction of questionnaire-based measures of theoretical constructs and methods for construing and validating such measures, which makes this reviewer think that the paper could have been more appropriately submitted to a statistical journal.

Since the authors have submitted the manuscript to a ‘general’ journal, this reviewer assumes that the authors wish to target a general audience with an interest in leadership theory and/or psychometric testing. If this is correct, the authors should consider presenting the analyses in less technical terms and elaborate more on the content and implications of the analyses.

Having presented these two general comments, the authors must be commended for having prepared a very interesting manuscript with interesting ramifications for the assessment of leaderships behaviors and for the psychometric assessment of such measures.

Reviewer #2: General concerns:

- In my opinion, the manuscript seems too extensive considering the design used. In this line, some ideas are mentioned in several paragrphs. For example, the nine first-order factors, three second-order factors, and one global MLQ-5X factor. In addition, relevant information it is not described. For example procedure, software, participants, instrument.

Major concerns:

- The sample is very small (n = 129). Although the authors address this fact in the limitations section, this can be the main issue in the validity of the described results. Yet, I leave it to the decision of the editor of the journal.

Additional concerns:

- The qualitative section, although interesting, its for me extensive and confusing. I suggest a table that summarises the comments for each item.

- There is no mention of the software used.

- It is not clear why the authors do not use additional fit indexes (e.g., CFI, TLI, IFI) and perform reliability analyses.

- It is not clear to me which questionnaire the participants answered (English, Spanish or Catalan version).

- The information about the participants is not described.

- It is not clear why the authors refer to the quasi-experimental design. Strictly speaking, the authors (for the manuscript) use a cross-sectional self-reporting design.

- The authors suggest replicating their study in different organisations, cultures and countries. However, they do not provide relevant information for doing so.

All the best

6. PLOS authors have the option to publish the peer review history of their article (what does this mean?). If published, this will include your full peer review and any attached files.

Reviewer #1: No

Reviewer #2: No

---

## [Author Response · Author response to Decision Letter 0]

29 May 2021

Response to Reviewer(s)

Manuscript ID PONE-D-20-05815:

MEASURING LEADERSHIP: An Assessment of the Multifactor Leadership Questionnaire

Dear editor, Rodrigo Ferrer, 

We thank to you as editor and the reviewers for supporting the paper and for the very constructive and helpful suggestions for improvement. We have attempted to address all the editor’s and the reviewers’ concerns, as detailed below, and have endeavored to provide a clearer exposition of the issues and the actions taken in the new draft of the manuscript. Key is that we invested in collecting fresh data from a second sample. As a consequence, we systematically re-wrote the paper to give center stage to (the comparison of) the findings, interpretations end conclusions associated with the analyses of both datasets. These are clearly marked in the additional file labeled “Revised Manuscript with Track Changes” (major changes are also marked in yellow). The end result is, we believe, a much stronger paper.

Editor comment 1 

I think the most interesting analysis the article has is that of content review, but without presenting the items, it is impossible to appreciate their value and make them useful beyond people who are broadly familiar with the MLQ (which would make the article relevant for a leading journal, but not for a broad audience such as PlosOne). 

Reply 

We completely agree with this comment. Due to copyright issues, it is not possible to publish the exact content of all the MLQ items. However, we have included a content analysis section in which we have described the measurement items without including the actual sentences. This new long section is titled “Content analysis” (pp. 15 to 20).

Editor comment 2 

Additionally, some of the psychometric analyses used are inadequate (or, at least, their adequacy has not been made explicit), e.g. the estimation method used is inadequate for ordinal variables, it is not specified whether the polychoric correlation matrix is used, relevant fit indicators are not reported, principal components should be used for the formative model and not CFA, among other things that can be remedied. 

Reply:

We agree with the editor that the ordinal nature should be mentioned, with the consequence that the polychoric correlation matrix should be used. We did actually decide to not mention this issue in the original draft because, although it leads to better SE, the estimates were pretty close, as was the fit of the model. Moreover, the matrix to be analyzed was not positive definite. We now solved with the ridge option (adding a constant 0.1 to the main diagonal of the polychoric correlation matrix). In the revised draft, we refer to robustness analyses with the polychoric matrix throughout the manuscript, sticking to the analyses with the Pearson correlation matrix as our main anchor for comparison (as extant work, albeit wrongly, uses Pearson). 

About using PCA with formative indicators, which is a tradition coming from Partial Least Squares, we disagree with this approach because it is based on correlations among items that, because they are not reflective, are not necessarily correlated. We did choose the more widely used strategy based on as parcels – i.e., summated rating scales. The strategy is also proposed for dealing with SEM with small sample sizes (see Bisbe, J., Coenders, G., Saris, W.E. & Batista-Foguet, J.M. (2006). Correcting measurement error bias in interaction models with small samples. Metodološki zvezki, 3(2), 267-288). Related, we also addressed to Reviewer 2’s comments and suggestions regarding the use of SEM. Regarding the explicit mentioning of other global fit indices in the diagnostic stage, we wanted to avoid what Kline (2010) termed “global fit indices tunnel vision”, which might lead us to focus on indices of overall model fit and to ignore more detailed diagnostic indicators.

 Actually, an important message of our paper regards the insensitivity of the global fit indices usually used for assessing research based on SEM. On page 18, we refer to the unidimensional factor solution test of the four items attached to each of the nine MLQ factors, mentioning that only a few of them would have been rejected, implying that many earlier studies wrongly continued on the basis of the (implicit) assumption that the nine MLQ factors are single-factor analysis models. We then point out that most of the global fit indices are associated with different levels of sensitivity for different misspecifications of the model and “For this reason, the usual testing procedure must be combined with a sensitivity analysis of the test statistics for all possible misspecifications (Saris et al. 2009).” Hence, besides the correlation matrix among the four items within each dimension (i.e., their loadings), we propose to consider not only the usual global fit indices and the attached significance tests, but also the power of the test as well as the content analysis. This diagnosis process clearly suggests that the one-factor solution underlying the nine MLQ factors is unlikely to work in many of the nine dimensions. Therefore, we followed the strategy proposed by Saris, Satorra, and Van der Veld (2009) by going through a set of more detailed diagnosis indicators, and focusing more on the detection of misspecification errors rather than solely on global fit, while also considering the power of each test.

 The main issue we have detected in MLQ is due to the specification of the nine dimensions as single-factor analysis models. And, in addition, the three particular subdimensions of the transactional leadership style barely correlate, even show negatively correlations, and are often more correlated with LF (associated with the transformational leadership style) than among themselves. The exception is CR. As mentioned in our revision, this factor correlates much more with the transformational leadership subdimensions.

Editor comment 3 & major concern Reviewer 2

- However, the major restriction to carry out these analyses is the limited sample size, so that any structure could be the product of mere chance, and it is necessary to increase it or provide other guarantees that give greater certainty in the results. 

- The sample is very small (n = 129). Although the authors address this fact in the limitations section, this can be the main issue in the validity of the described results.

Reply: 

We understand the reservations that both the editor and reviewer 2 express regarding the small sample size used in our study. To remedy this limitation, we decided to collect more data by adding another sample. As described in the methods section, we have included 300 more responses from an online survey of US citizens, using the online platform Prolific. The new version of the manuscript includes the analysis of the two study samples, and discusses the differences and similarities between and across both samples.

Reviewer 1 comment 1 

The authors provide an interesting and highly relevant discussion of the face validity of the items of the MLQ that are used to operationalize subscales of the MLQ. The authors state that they cannot present the individual items of the MLQ due to copyright issues. This is a big shame, as it is quite difficult to follow the discussion of the face validity of the items. Therefore, the authors should decide whether to 1) omit the discussion of the face validity, as it makes little sense in the way it is presented in the current version of the manuscript, 2) provide a more thorough description of the contents of the items (e.g. in a table), so that the reader is actually enabled to follow the discussion or 3) to present the items and deal with the copyright issues. As it stands now, this reviewer does not find that the basis of the discussion of the face validity of the MLQ satisfactory.

Reply: 

We thank the reviewer’s suggestion as to how to deal with the content validity of the MLQ. As we believe that the actual writing of the items is important to understand the limitations of this measure, we have included a new section titled “Content analysis”, in which we have carefully expanded our description of the items without providing the exact items, so that we can comment on their meaning without violating any copyrights. As mentioned above, this new section can be found on pp. 15 to 20.

Reviewer 1 comment 2

Another general comment is that this reviewer is a bit uncertain about the target group of the study, which leaves me wondering, why it was submitted to a general journal as PlosOne. First, the paper deals with a very specific problem regarding the measurement of leadership behaviors, which makes this reviewer think that the paper could have been more appropriately submitted to a leaderships journal. Second, the paper has its’ focus on very specific issues regarding construction of questionnaire-based measures of theoretical constructs and methods for construing and validating such measures, which makes this reviewer think that the paper could have been more appropriately submitted to a statistical journal. Since the authors have submitted the manuscript to a ‘general’ journal, this reviewer assumes that the authors wish to target a general audience with an interest in leadership theory and/or psychometric testing. If this is correct, the authors should consider presenting the analyses in less technical terms and elaborate more on the content and implications of the analyses.

Reply: 

Thank you very much for this interesting reflection. We decided to target a general journal as PLoSONE because the MLQ has been used widely—and still is so – across multiple disciplines (e.g., in business, public administration, and psychology). It is arguably the most frequently used measurement tool to assess leadership. Because of that, our aim is for our critique to reach a large and diverse audience, not simply those that work on leadership in the HR and OB fields. Furthermore, in line with the reviewer’s comments, we have edited the whole manuscript to ensure that it would be appealing to a general audience, with MLQ as a case standing for a much more widespread issue of problematic scale psychometrics, and not only to individuals with high statistical expertise. Some more technical issues are commented upon in footnotes.

Reviewer 2 additional concerns

- The qualitative section, although interesting, its for me extensive and confusing. I suggest a table that summarises the comments for each item.

Reply: Upon reading the qualitative section, we realized that it was rather difficult to follow. This has been an issue raised by the other reviewer, too. Hence, we have extensively rewritten this section to clarify the content analysis of the MLQ. We have opted to rewrite the section instead of including a table because we believe that this will help readers to follow our argumentation as to why the content validity of some dimensions should be questioned. This new section can be found on pp. 15 to 20.

- There is no mention of the software used.

Reply: Our statistical analysis was performed using LISREL8.80. We have included this information on p. 20 of the manuscript. Note that we re-ran the analyses with LISREL10.2 version. Although some fit indices vary somewhat, this does not affect the interpretations or conclusions. So, we decided to stick to the results from LISREL8.80.

- It is not clear why the authors do not use additional fit indices (e.g., CFI, TLI, IFI) and perform reliability analyses

Reply: In our reply to the editor’s second comment, we offer our response to this issue. Here, we would just like to add that, as we say in the text (p. 20), “Table 1 shows that had we tested the unidimensionality of the four items attached to each MLQ factor, only a few of them would have been rejected, implying that many earlier studies wrongly continued on the basis of the (implicit) assumption that the MLQ is a reflective model.” Moreover, we have added this footnote: “Notice that the 0-4 (or 1-5) answer modality used in MLQ actually implies ordinal properties of the data gathered. Therefore, the Polychoric correlation matrix should be the one to be analyzed. However, since the standard practice is to use the Pearson covariance correlation matrix for the analysis, we provide the results of this second approach, and make explicit comments when we observed relevant differences between both analyses. In general, we could say that the appropriate Polychoric correlations lead to a better fit of the CFA model, as well as higher loading estimates. However, differences in the US population sample are lower than in the Catalan Police sample. In the latter, the model actually lost even more sensitivity to detect unidimensionality misspecifications.”

- It is not clear to me which questionnaire the participants answered (English, Spanish or Catalan version).

- The information about the participants is not described.

- It is not clear why the authors refer to the quasi-experimental design. Strictly speaking, the authors (for the manuscript) use a cross-sectional self-reporting design. 

- The authors suggest replicating their study in different organizations, cultures and countries. However, they do not provide relevant information for doing so.

Reply: We thank the reviewer for these comments. Below, we answer each of them.

-We missed to mention that we did used Catalan versions in our police officer’s study, and English in the US one. We have included this information in the new version.

- The reviewer is right. We have added demographic information of our respondents in the amended version.

- This research was originally designed to assess the effect of a transformational leadership training in the Catalan police force. In that research, we proposed a quasi-experimental single-group pre-post design with non-equivalent dependent variables, for assessing the effect of the training delivered. However, for the current paper, we consider the post test data, as the reviewer rightly pointed out. Hence, we have deleted this reference to the original research design. 

- Regarding the reproducibility of our results, we have made our databases available, for both samples. The only restriction is that we cannot publish the exact words used n the MLQ questionnaire because of copyright issues. The company that owns the MLQ rights is MindGarden, and they have a very strict copyright policy regarding the MLQ questionnaire. However, we believe that with the databases that we provide any author could reproduce the analysis that we have performed.

We want to thanks the editor and the two reviewers for their incredibly helpful comments. Their suggestions have really helped us to enhance this manuscript.

---

## [Editor Report · Decision Letter 1]

28 Jun 2021

MEASURING LEADERSHIP: An Assessment of the Multifactor Leadership Questionnaire

PONE-D-20-05815R1

Dear Dr. Batista-Foguet,

We’re pleased to inform you that your manuscript has been judged scientifically suitable for publication and will be formally accepted for publication once it meets all outstanding technical requirements.

Kind regards,

Rodrigo Ferrer, Ph.D.

Academic Editor

PLOS ONE

Additional Editor Comments (optional):

Dear Authors, first of all I am apologizing for the huge delay in the review, but I have been facing health issues and a multitude of new responsibilities that have led me to fail. Regarding your article, it seems to me that you addressed all the suggested improvements and, those that you did not implement, have a reasonable foundation, so I endorse their publication. Additionally, I would like to compliment you for looking at the formative models and providing a discussion that, despite being very necessary, is very rare.

Greetings and best wishes
---

## [Editor Report · Acceptance letter]

14 Jul 2021

PONE-D-20-05815R1 

MEASURING LEADERSHIP An Assessment of the Multifactor Leadership Questionnaire 

Dear Dr. Batista-Foguet:

I'm pleased to inform you that your manuscript has been deemed suitable for publication in PLOS ONE. Congratulations! Your manuscript is now with our production department. 

Kind regards, 

on behalf of

Dr. Rodrigo Ferrer 

Academic Editor

PLOS ONE